# Generalizable Dynamic Radiance Field in Egocentric View

## Abstract

We present a novel framework for generalizable[1] dynamic radiance field in egocentric view. Our approach can predict a 3D representation of the physical world at a given time based on a monocular video without test-time training. To this end, we use a contracted triplane as the 3D representation of physical world in an egocentric view at a specific time. To update the explicit 3D representation, we propose a 4D-aware transformer module to aggregate features from monocular videos. Besides, we also introduce a temporal-based 3D constraint to achieve better multiview consistency. In addition, we train the proposed model with large-scale monocular videos in a self-supervised manner. Our model achieves top results in novel view synthesis on dynamic scene datasets, demonstrating its strong understanding of 4D physical world. Besides, our model also shows the superior generalizability to unseen scenarios. Furthermore, we find that our approach emerges capabilities for geometry and semantic learning. We hope our approach can provide preliminary understanding of the physical world in first-person view and help ease future research in computer vision, computer graphics and robotics.

## 1 Introduction

As humans, we perceive and interact with physical world in a first-person perspective. Meanwhile, we endow physical world with semantic meaning by our behaviors during interaction process. Besides, we can generalize our knowledge about 4D physical world to various environments. To build human intelligence in physical world, it is reasonable to model dynamic world in egocentric view, which not only establishes physical world perception but also simulates human behaviors. To this end, machine learning models should bear two characteristics: generalized dynamic scene modeling and egocentric learning mode.

Dynamic scene modeling (Li et al., 2023; Park et al., 2021; Tian et al., 2023; Yang et al., 2023a; Zhou et al., 2023; Yang et al., 2023b) has achieved great improvements with the advancements of neural rendering (Srinivasan et al., 2020; Kerbl et al., 2023). However, many methods (Li et al., 2021b; Gao et al., 2021; Wang et al., 2022; Li et al., 2022; 2023) require specific-scene optimization, which do not have generalizability. Although some methods (Tian et al., 2023; Zhao et al., 2024) leverage generalized prior inputs for dynamic scene modeling, these works still lack generalizability to unseen scenarios. Overall, the generalizability issue of existing methods is derived from their object-centric modeling. In contrast, ego-centric modeling is independent of specific scenes, thus it is a reasonable alternative to achieve generalized dynamic scene modeling without extra priors.

In this paper, we propose a framework for generalizable dynamic radiance field in first-person view. The proposed method obtains the ability of cross-scene dynamic view synthesis in a self-supervised and data-driven manner. Specifically, we use an explicit 3D representation to represent physical world in egocentric view at target time $t_1$. For the explicit 3D representation, we adopt a contracted triplane to represent the unbounded scenes. Given a sequence of egocentric views at time $\{t : t \neq t_1\}$, we propose a 4D-aware transformer to update the triplane features. Using the updated triplane, we render the egocentric view with volume rendering at time $t_1$. To achieve better multiview consistency, we introduce a temporal-based 3D constraint, which renders two views that are temporally distant given the same sequence of source views. Also, our approach is trained on large-scale monocular videos in

---

[1]The "generalizability" is defined as not requiring optimization, fitting, training, or fine-tuning on test scenes.

egocentric view. In addition, our method can perform dynamic novel view synthesis, showing its 4D understanding abilities. Our model also showcases its generalizability on unseen scenarios.

To validate the effectiveness of our method, we conduct experiments on the NVIDIA Dynamic Scenes, EPIC Fields and Plenoptic Video datasets to assess the ability of novel view synthesis. Subsequently, we test the generalization capability of our model on the completely unseen scenes, highlighting the strong ability to transfer to novel scenes. In our ablation study, we thoroughly analyze the effects of the core components of our model, highlighting their strengths and contributions to overall performance. Moreover, we find emergent capabilities of our model, including geometry and semantic learning.

To sum up, our approach is the first work for generalizable dynamic radiance field in first-person view. We validate our method by conducting extensive experiments on various settings, especially generalization on unseen scenarios. In addition, our method can be viewed as a potential 4D world compressor, representing real world in egocentric view and simulating human behaviors simultaneously.

## 2 RELATED WORK

### 2.1 RADIANCE FIELD RENDERING

Radiance field rendering has recently obtained a remarkable accomplishment. Neural Radiance Fields (NeRFs) (Mildenhall et al., 2021) use MLPs to represent scenes and volume rendering to render high-quality images at novel viewpoints. The success of NeRF has resulted in numerous subsequent works that address its shortcomings (Barron et al., 2022; Chan et al., 2022; Müller et al., 2022; Yu et al., 2021; Chen et al., 2022) and expand its applications (Poole et al., 2022; Wang et al., 2024; Hong et al., 2023). Mip-NeRF360 (Barron et al., 2022) demonstrates impressive view synthesis results on unbounded scenes. EG3D (Chan et al., 2022) and InstantNGP (Müller et al., 2022) use a triplane or a hash grid to accelerate computation, separately. However, these methods inevitably face a trade-off between speed and quality. To solve this obstacle, 3D Gaussian Splatting (Kerbl et al., 2023) is proposed. It takes 3D Gaussians as a scene representation, projects them into 2D via a rasterization mechanism and renders image as NeRF. For its real-time speed and high quality, many follow-up works (Yan et al., 2023; Fu et al., 2023a; Szymanowicz et al., 2023; Ling et al., 2023) are rapidly emerging. (Yan et al., 2023) represents a scene with multi-scale 3D Gaussians to address aliasing effect. CF-3DGS (Fu et al., 2023a) performs novel view synthesis without any SfM preprocessing by leveraging the explicit point cloud and the continuity of the input video stream. The existing neural rendering methods mainly focus on scene reconstruction and novel view synthesis for a specific scene, while our method aims at generalizable dynamic scene synthesis task.

### 2.2 DYNAMIC NOVEL VIEW SYNTHESIS

Rendering (reconstructing) dynamic 3D scenes is critical for many applications, from AR/VR to autonomous driving (Yang et al., 2023a; Zhou et al., 2023; Yang et al., 2023b). Many works (Park et al., 2020; Pumarola et al., 2021; Wang et al., 2022; Li et al., 2022; Fridovich-Keil et al., 2023; Cao & Johnson, 2023) on dynamic scene synthesis requires multi-view input videos. D-nerf (Park et al., 2020) and Nerfies (Pumarola et al., 2021) represent scenes by mapping each observed point into a canonical scene representation via a volumetric deformation field. However, these methods are limited to object-centric scenes with relatively small object motion. DyNeRF (Li et al., 2022) and K-Planes (Fridovich-Keil et al., 2023) compress dynamic scenes into implicit or explicit NeRF and render these scenes with position, view direction and time conditions. To break the constraint of multi-view data, some approaches (Li et al., 2023; Tian et al., 2023; Zhao et al., 2024) are proposed to represent dynamic scenes from monocular videos. DynIBaR (Li et al., 2023) synthesizes novel image by aggregating image features from nearby views in a scene motion-aware manner. But DynIBaR (Li et al., 2023) focuses on specific-scene optimization and can not generalize to unseen scenarios. The most relevant works, MonoNeRF (Tian et al., 2023) and PGDVS (Zhao et al., 2024) still need scene-specific optimization or finetuning when transferring to unseen scenes. Also, they rely on extra priors, especially semantic masks of foreground objects. Unlike existing works, our method implement a generalizable dynamic scene synthesis only with monocular videos.

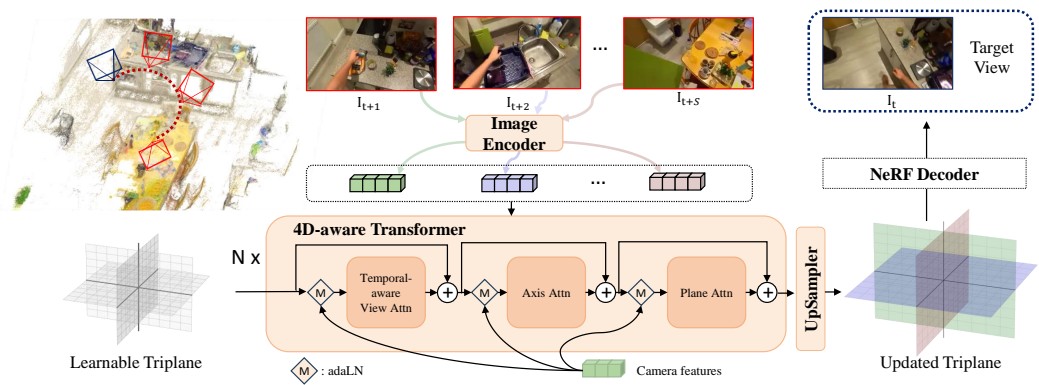

Figure 1: **Overview of our method.** The process starts with an image encoder that processes the source view of a scene, generating 2D image features as a prior. Then a 4D-aware transformer takes these image features to update a learnable triplane. An upsampler subsequently enlarges the triplane. Finally, the decoder retrieves 3D point features from the triplane along the view direction and utilizes these features to compute RGB values and densities for volumetric rendering. *The FFN module in the transformer is hidden in the figure.*

## 3 METHOD

In this work, our method aims to learn generalizable dynamic scene representation in first-person view from monocular videos. In particular, given a monocular video comprising S frames $\{\mathbf{I}_t\}_{t=1}^S$ and corresponding camera parameters $\{\mathbf{P}_t\}_{t=1}^S$, our model synthesizes a novel view of dynamic scene at target time $t_1$ without per-scene training. As shown in Figure 1, we firstly use an image encoder to extract image features from a monocular video, then take these image features as condition to update a learnable explicit 3D representation via a 4D-aware transformer, finally upsample the triplane and render novel views using volume rendering. In addition, we also introduce a temporal-based 3D constraint to improve multiview consistency.

### 3.1 IMAGE ENCODER

Given a sequence of source views $\{\mathbf{I}_t\}_{t=1}^S$, we use a hybrid neural network to extract image features $\{\mathbf{F}_t\}_{t=1}^S$. For efficiency and simplicity, the hybrid network consists of a resnet-like backbone and a self-attention layer. The resnet-like backbone downsamples the spatial dimension of the input images by a factor of 8. The self-attention layer is used to enhance the long-context ability of image features, benefiting the motion-aware feature aggregation in view-attention module 3.2.2.

### 3.2 SCENE GENERATOR

#### 3.2.1 CONTRACTED EGO-TRIPLANE REPRESENTATION

To represent a scene in egocentric view, we take camera center as world origin. Note that in our paper, egocentric view is only a modeling approach. It takes observer as world origin to model dynamic scenes. For each video frame, we use camera center as world origin. Thus, under ego-view modeling, all videos can be taken as egocentric videos.

For 3D representation, we adopt a learnable triplane and set camera center as triplane origin. The learnable triplane $\mathbf{T}$ contains three axis-aligned feature planes $\mathbf{T}_{xy}$, $\mathbf{T}_{yz}$, and $\mathbf{T}_{xz}$. The dimension of each plane is $H \times W \times D$, where $H \times W$ is the spatial dimension and $D$ is the feature channels. Since our egocentric representation is for scenes that unbounded in all directions, we need a transformation between the unbounded ego coordinates and bounded triplane coordinates. To this end, we adopt a

Figure 2: **Temporal-aware View-Attention and Axis-Attention modules in Transformer. (a) Temporal-aware View-Attention Module**: At target time $t_1$, 3D virtual points are uniformly sampled within the triplane. For a specific point $\mathbf{x}_{i,j,k}$, it is projected along the three axes onto the triplane features $\mathbf{T}_{xy}$, $\mathbf{T}_{yz}$, and $\mathbf{T}_{xz}$ to derive the 3D query feature $\mathbf{q}_{i,j,k}$. Simultaneously, the point $\mathbf{x}_{i,j,k}$ is mapped onto image feature maps to obtain epipolar features $\{\mathbf{f}_{t=1}^{S}\}$. The temporal-aware view-attention module integrates these epipolar features from $S$ image views according to times $\{t\}_{t=1}^{S}$, resulting in an updated 3D query feature $\hat{\mathbf{q}}_{i,j,k}$. **(b) Axis-Attention Module**: The triplane feature $\mathbf{p}_{i,j}$ at pixel $(i,j)$ located in plane $\mathbf{T}_{xy}$ associates with point features along the z-axis $\{\hat{\mathbf{q}}_{i,j,k}\}_{k=1}^{L}$. The axis-attention module aggregates these point features to yield a new triplane feature $\hat{\mathbf{p}}_{i,j}$.

non-linear contraction function from mip-NeRF 360 (Barron et al., 2022):

$$\mathcal{C}(\mathbf{x}) = \begin{cases} \mathbf{x} & \|\mathbf{x}\| \leq 1 \\ \left(2 - \frac{1}{\|\mathbf{x}\|}\right)\left(\frac{\mathbf{x}}{\|\mathbf{x}\|}\right) & \|\mathbf{x}\| > 1 \end{cases} \tag{1}$$

where x is 3D point in ego coordinates, $\mathcal{C}(\mathbf{x})$ is the point in triplane coordinates.

### 3.2.2 4D-AWARE TRANSFORMER

With the initial triplane representation, we propose a 4D-aware transformer to update the triplane features. The 4D-aware transformer include three core components: temporal-ware view-attention module, axis-attention module, plane-attention module.

**Temporal-aware View-Attention Module** The temporal-aware view-attention module aims to aggregate image features $\{\mathbf{F}_t\}_{t=1}^{S}$ from source views in a motion-aware manner, as shown in Figure 2 (a). Concretely, we uniformly sample 3D virtual points in triplane space. We denotes these virtual point set as $\{\mathbf{x}_{i,j,k}\}_{i=1}^{M}\,_{j=1}^{N}\,_{k=1}^{L}$, where $i$, $j$ and $k$ represent x, y and z axes, $M$, $N$ and $L$ is the point number in each axis, respectively. In temporal-aware view attention module, we take virtual point feature as query and its corresponding epipolar point features in $\{\mathbf{I_t}\}_{t=1}^{S}$ as key and value. Note that we also take time $t_1$ and other times $\{t\}_{t=1}^{S}$ as query and key to compute attention scores. For a virtual point $\mathbf{x}_{i,j,k}$, we compute its $\mathbf{q}_{i,j,k}$ by projecting it onto three planes of the triplane, querying the corresponding point features and concating these features. To obtain epipolar point features $\{\mathbf{f}_t\}_{t=1}^{S}$, we project $\mathbf{x}_{i,j,k}$ onto the t-th image plane by applying camera pose $\mathbf{P}_t$ and compute the feature via bilinear interpolation on the image feature grids. Thus, the new virtual point feature $\hat{\mathbf{q}}_{i,j,k}$ is computed as:

$$\hat{\mathbf{q}}_{i,j,k} = CrossAttn(\mathbf{q}_{i,j,k}, t_1, \{\mathbf{f}_t\}_{t=1}^{S}, \{t\}_{t=1}^{S}) + \mathbf{q}_{i,j,k} \tag{2}$$

Previous works (Li et al., 2023; Tian et al., 2023) learn motion trajectory for 3D point $\mathbf{x}$ and use estimated motion trajectory of $\mathbf{x}$ to aggregate feature for dynamic content. Meanwhile, these works leverage epipolar geometry as an inductive bias to aggregate feature for static content. The whole aggregation process requires motion/semantic segmentation to determine dynamic content and static content. In contrast, our temporal-aware view-attention module can determine dynamic and static contents implicitly. This is validated by the temporal-aware similarities between the 3D virtual point feature $\mathbf{q}_{i,j,k}$ and its epipolar point features $\{\mathbf{f}_t\}_{t=1}^{S}$, see Appendix A.2. For virtual point features of dynamic content, they contain much noise, since epipolar constraint is invalid in dynamic scenes. To solve this issue, we propose axis-attention module and plane-attention module to refine virtual point features.

**Axis-Attention Module** The cost of directly applying self-attention to refine virtual point features is huge, thus, we instead implement it by refine triplane features and introduce axis-attention module

to project new virtual point features onto triplane. In axis-attention module, we take triplane features as query and its corresponding virtual point features as key and value. As shown in Figure 2 (b), take plane $\mathbf{T}_{xy}$ as an example, a sequence of virtual point features $\{\hat{\mathbf{q}}_{i,j,k}\}_{k=1}^{L}$ along $z$ axis corresponds to triplane feature $\mathbf{p}_{i,j}$ on $\mathbf{T}_{xy}$. In axis-attention module, we also include a position bias (3D position embedding) to each head in computing similarity. Finally, we compute new triplane feature $\hat{\mathbf{p}}_{i,j}$ on plane $\mathbf{T}_{xy}$ with cross-attention, formulated as follows:

$$\hat{\mathbf{p}}_{i,j} = CrossAttn(\mathbf{p}_{i,j}, \{\hat{\mathbf{q}}_{i,j,k}\}_{k=1}^{L}) + \mathbf{p}_{i,j} \tag{3}$$

**Plane-Attention Module** In temporal-aware view-attention module, the aggregated features are not good enough to model dynamic contents, since epipolar constraint is invalid in dynamic scenes. Thus, we build a plane-attention module to leverage 3D-related information. Like (Cao et al., 2023), our plane-attention module includes self- and cross-plane attention. The self-plane attention aims to enhance the semantic information of individual planes while the cross-plane attention concentrates on building generalized 3D prior knowledge across different planes. Therefore, we can leverage learned semantic information and 3D prior knowledge to resolve the feature error from temporal-aware view-attention module.

Specifically, the aim of self-plane attention is to update each plane feature by aggregating features from intra-plane. In cross-plane attention, each plane feature take itself as query and other planes as key and value. In plane-attention module, we also add a position bias to each head in computing similarity. Note that our contracted triplane is used to represent unbounded scenes. In general, we need to embed the infinite position, which is hard to implement. To avoid it, we take learnable position embedding to represent position bias.

**Camera Feature** The training dataset comprises images with a broad range of focal lengths, causing scale ambiguity. To address it, we take camera intrinsic matrix as an inductive bias. We construct a camera feature $\mathbf{c} \in \mathbb{R}^{16}$ for each target view by flattening its 4-by-4 camera intrinsic matrix. Then we embed feature $\mathbf{c}$ by mapping it into a higher-dimensional space via a sinusoidal function $\gamma(\cdot)$ and projecting it into input dimension via a multi-layer perceptron (MLP). Drawing inspiration from DiT (Peebles & Xie, 2023), we incorporate an adaptive layer normalization (adaLN) within our feature attention block to effectively constrain the inputs of each attention block based on camera features.

### 3.3 SCENE DECODER

**Upsampler** For high performance, we use a trainable deconvolution layer to upscale the triplane embeddings $\mathbf{T}_{xy}, \mathbf{T}_{yz}$ and $\mathbf{T}_{xz}$ extracted from the transformer. After the upsampling, we obtain the final triplane for volume rendering.

**NeRF Decoder** We adopt NeRF as decoder to predict color RGB and density $\sigma$ based on the 3D point feature extracted from the triplane. Initially, we normalize the 3D position using the contraction function described in Section 3.2.1 and project it onto three planes. Subsequently, we concatenate the features from these planes to form the final feature vector. This vector is then decoded into color RGB and density $\sigma$ using a lightweight MLP. In addition, we normalize the values of 'near' and 'far' for all scenarios.

### 3.4 TRAINING

**Temporal-based 3D Constraint** A strong multi-view constraint, which generates multiple views simultaneously from the same scene, is a common strategy to ensure 3D consistency. However, given the limitations of monocular video, we employ a temporal-based 3D constraint by rendering two views that are temporally distant conditioned on the same sequence. Specifically, given a sequence of egocentric views $\{\mathbf{I}_t\}_{t=1}^{S}$, we select two frames that are $S$ frames apart as the target views, leveraging the significant disparity between these time-distant views to enforce the constraint.

**Training Strategy** We adopt a cost-effective training strategy by starting with lower-resolution input images. Initially, we pretrain our model on $128 \times 72$ images until convergence. Then, we finetune it using $512 \times 288$ images, significantly reducing computational costs while still achieving superior quality compared to direct high-resolution training with equivalent computational resources.

Table 1: **Results on NVIDIA Dynamic Scenes.** "General" refers to generalizability [1] of the models. "Priors" means using pre-trained priors, such as depth and semantic segmentation. [†] specifies the generalized variant of PGDVS with input depth from ZoeDepth (Bhat et al., 2023).

| Model | General | Priors | Full Image | | | Dynamic Area | | | Static Area | | |
|---|---|---|---|---|---|---|---|---|---|---|---|
| | | | PSNR↑ | SSIM↑ | LPIPS↓ | PSNR↑ | SSIM↑ | LPIPS↓ | PSNR↑ | SSIM↑ | LPIPS↓ |
| DVS | ✗ | ✓ | 27.96 | 0.912 | 81.93 | 22.59 | 0.777 | 144.7 | 29.83 | 0.930 | 72.74 |
| NSFF | ✗ | ✓ | 29.35 | 0.934 | 62.11 | 23.14 | 0.784 | 158.8 | 32.06 | 0.956 | 46.73 |
| DynIBaR | ✗ | ✓ | 29.08 | 0.952 | 31.20 | 24.12 | 0.823 | 62.48 | 31.68 | 0.971 | 25.81 |
| MonoNeRF[2] | ✗ | ✓ | 22.06 | 0.751 | 18.30 | 15.40 | 0.307 | 68.50 | 25.03 | 0.803 | 14.10 |
| PGDVS | ✗ | ✓ | 26.15 | 0.922 | 64.29 | 20.64 | 0.744 | 104.4 | 28.34 | 0.947 | 57.74 |
| PGDVS[†] | ✓ | ✓ | 21.15 | 0.814 | 142.3 | 15.93 | 0.479 | 233.5 | 23.36 | 0.854 | 129.9 |
| Ours | ✓ | ✗ | 22.43 | 0.706 | 16.29 | 18.64 | 0.652 | 33.04 | 24.03 | 0.724 | 15.79 |

Table 2: **Comparison of novel view synthesis task on RealEstate10K.** "$n$" is the number of frames between the source and target frames.

| Model | LPIPS↓ | | | SSIM↑ | | | PSNR↑ | | |
|---|---|---|---|---|---|---|---|---|---|
| | $n = 5$ | $n = 10$ | $n =$ rand | $n = 5$ | $n = 10$ | $n =$ rand | $n = 5$ | $n = 10$ | $n =$ rand |
| MINE | 8.96 | 12.8 | 15.62 | 0.8974 | 0.8500 | 0.8219 | 28.39 | 25.71 | 24.50 |
| MonoNeRF-static | 14.3 | - | - | 0.8600 | - | - | 26.68 | - | - |
| Ours | 4.52 | 7.01 | 9.21 | 0.8231 | 0.7700 | 0.7521 | 25.73 | 23.73 | 21.85 |

**Training Objective**   To mitigate the high cost of rendering full-resolution images for volume rendering, we randomly select $64 \times 64$ and $128 \times 128$ patches from target images with resolutions of $128 \times 72$ and $512 \times 288$, respectively, for loss supervision. We evaluate the visual accuracy of our renderings compared to ground-truth (GT) images using three types of losses: an L2 reconstruction loss $\mathcal{L}_{\text{recon}}$, a perceptual loss $\mathcal{L}_{\text{lpips}}$, and a structural similarity loss $\mathcal{L}_{\text{ssim}}$. Additionally, to address the issue of semi-transparent clouds, we apply a regularization term $\mathcal{L}_{\text{dist}}$, inspired by the distortion loss in Mip-NeRF360 (Barron et al., 2022). The overall training loss function is formulated as follows:

$$\mathcal{L} = \mathcal{L}_{\text{recon}} + \lambda_{\text{lpips}}\mathcal{L}_{\text{lpips}} + \lambda_{\text{ssim}}\mathcal{L}_{\text{ssim}} + \lambda_{\text{dist}}\mathcal{L}_{\text{dist}} \tag{4}$$

where $\lambda_{\text{lpips}}$, $\lambda_{\text{ssim}}$ and $\lambda_{\text{dist}}$ are the scale to balance the perceptual, structural similarity and distortion regularization respectively, which is set to be 0.1, 0.1 and 0.01 in our experiments.

## 4   EXPERIMENTS

**Training Details**   We utilize the Adam optimizer with a learning rate of 0.0001, $\beta_1 = 0.9$, and $\beta_2 = 0.999$. Additionally, we adopt learning rate warm-up for the early stable training. Our model is trained on 32 NVIDIA A100 GPUs with a batch size of 128 for 1000 epochs at $128 \times 72$ and $512 \times 288$ resolutions respectively. In addition, for each input sequence, we define the triplane orientation using the camera direction of the middle frame in the source sequence.

**Datasets**   We conduct experiments on several datasets:

*NVIDIA Dynamic Scenes* (Yoon et al., 2020): The NVIDIA Dynamic Scenes dataset is a widely used benchmark for evaluating dynamic scene synthesis. It comprises eight dynamic scenes captured by a synchronized rig with 12 forward-facing cameras. We derive monocular videos following the setup in prior works (Li et al., 2023; VIDEO), ensuring that the resulting monocular videos cover most timesteps.

*EPIC Fields* (Tschernezki et al., 2024): The EPIC-KITCHENS is a comprehensive egocentric dataset. EPIC Fields extends EPIC-KITCHENS by including 3D camera poses. This augmentation

---

[2]Note that here we follow the data processing method of MonoNeRF to process Nvidia Dynamic Scenes dataset with all eight scenes and the whole frames. The training step is set to 40000, same as the default, and we evaluate novel view synthesis with the same settings as ours.

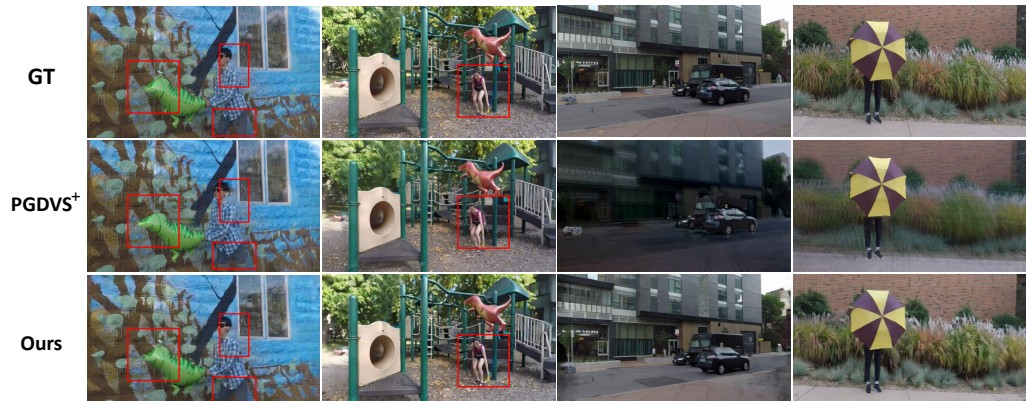

Figure 3: **Qualitative Comparison on the NVIDIA Dynamic Scenes Dataset.** Our method outperforms PGDVS[†] significantly in both dynamic and static content. For dynamic objects (first two columns), compared with PGDVS[†], our approach present accurate motion and avoid motion blur. For static scenes (last two columns), our method shows clear background, while PGDVS[†] produce blurred background due to limited depth priors.

reconstructs 96% of the videos in EPIC-KITCHENS, encompassing 19 million frames recorded over 99 hours in 45 kitchens. To minimize redundancy and skew while ensuring sufficient viewpoint coverage, we apply the frame filtering method from Tschernezki et al. (2024) to extract monocular videos.

*nuScenes* (Vora et al., 2020): The nuScenes dataset is a large-scale autonomous driving benchmark, comprising 1000 scenes, pre-split into training and test sets. Each sample includes RGB images from six cameras, providing a 360° horizontal field of view. For our experiments, we use only the three forward-facing camera views to extract monocular videos. We adopt the dataset split from Vora et al. (2020) for generalization testing.

*Plenoptic Video dataset* (Li et al., 2022): The Plenoptic Video dataset is a real-world dataset captured using a multi-view camera system consisting of 21 GoPro cameras operating at 30 FPS. Each scene in the dataset comprises 19 synchronized videos, each 10s in duration.

*RealEstate10K* (Zhou et al., 2018): The RealEstate10K dataset (Zhou et al., 2018) is a large-scale collection of walkthrough videos featuring both indoor and outdoor scenes. It comprises over 70,000 video sequences. Each sequence includes video frames along with their corresponding camera intrinsics and extrinsics.

*DAVIS* (Perazzi et al., 2016): The DAVIS dataset is a high-quality video object segmentation benchmark consisting of 50 video sequences with 24 FPS and Full HD resolution. We process video sequences using the same settings as prior work Zhao et al. (2024) and evaluate generalization ability on this dataset.

**Metrics** We use novel view synthesis task to validate dynamic scene modeling of our model. To evaluate the quality of novel view synthesis, we use Learned Perceptual Image Patch Similarity (LPIPS) (Zhang et al., 2018), Structural Similarity Index (SSIM) (Wang et al., 2004), and Peak Signal-to-Noise Ratio (PSNR).

## 4.1 NOVEL VIEW SYNTHESIS

In this section, we present both quantitative(Section 4.1.1) and qualitative results(Section 4.1.2) to demonstrate the effectiveness of our proposed method. We utilize six datasets, organized into seven collections: NVIDIA, EPIC, Plenoptic, RealEstate10K, nuScenes (train set), nuScenes (test set) and DAVIS. Our model is trained on scenes from EPIC, Plenoptic, and the nuScenes train set, and is subsequently evaluated on unseen scenes as detailed below.

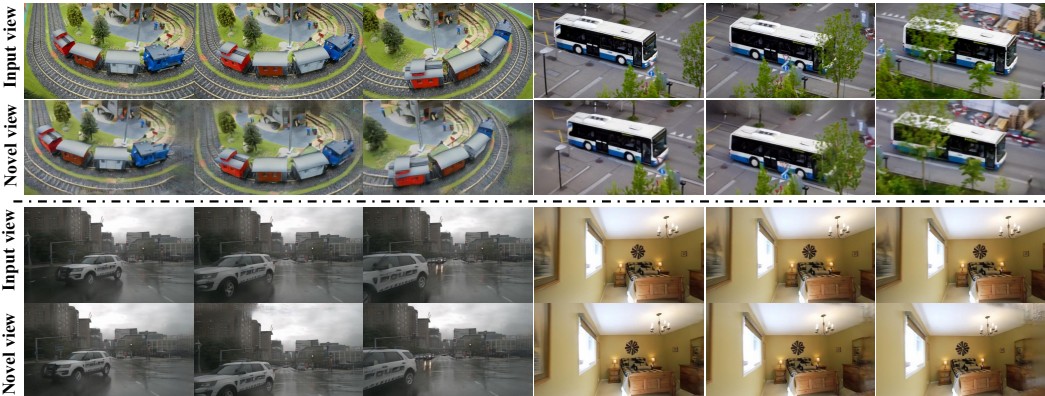

Figure 4: **Novel View Synthesis Across Diverse Datasets**. Our model produces high-quality novel views across diverse scenes, including indoor, outdoor, dynamic, and static settings. The top rows highlight results on the DAVIS dataset. The bottom rows reveal its performance in entirely new environments of the nuScenes and RealEstate10K datasets.

### 4.1.1 QUANTITATIVE EVALUATION

**Setup**  We conduct quantitative evaluations on the NVIDIA Dynamic Scenes and RealEstate10K datasets. To ensure fair novel view synthesis, the NVIDIA dataset is processed following the PDGVS setup. Each scene frame consists of 12 views, all views are used for testing. In RealEstate10K scenes, following the setup from MINE (Li et al., 2021a) and MonoNeRF-static (Fu et al., 2023b), we replicate the reference frame six times as source images. Novel views are selected either as 5 or 10 frames from the reference sequence or randomly from a range of 30 frames to provide more distinct views. For both datasets, the scenes were unseen during training, and we excluded scene-specific optimization to evaluate the generalization capability of our model.

**Results**  On the NVIDIA Dynamic Scenes dataset, we compare our method to the scene-specific approaches NSFF (Li et al., 2021b), DVS (Gao et al., 2021) and DynIBaR (Li et al., 2023), as well as the pseudo-generalized method MonoNeRF (Tian et al., 2023) and PGDVS (Zhao et al., 2024). For fair comparison, we set the baseline as completely generalized PGDVS[†] which utilizes depth input from ZoeDepth (Bhat et al., 2023). Our method outperforms both PGDVS[†] and MonoNeRF across all metrics for the dynamic components of the scenes, achieving a PSNR of 18.64 compared to 15.40 and 15.93, respectively. This highlights that our prior-free, self-supervised approach effectively learns semantics and motion, excelling at capturing long-term dynamics in dynamic scenes. For the static components, our method also surpasses PGDVS[†] and MonoNeRF in both PSNR and LPIPS metrics. Notably, our method significantly excels in the LPIPS metric, highlighting its enhanced scene synthesis capabilities. Although our approach lags behind scene-specific methods, our method has a potential performance improvement, since it is a data-driven method and can benefit from large-scale datasets.

For the RealEstate10K dataset, we evaluate our model against two single-image novel view synthesis methods, MINE (Li et al., 2021a) and MonoNeRF-static (Fu et al., 2023b). As illustrated in Table 2, our method performs on par with these state-of-the-art techniques across all test settings, despite the RealEstate10K domain being entirely excluded from our training data. Additionally, our model achieves superior performance on the LPIPS metric, underscoring its strong ability to adapt to new and unseen scenes.

### 4.1.2 QUALITATIVE EVALUATION

**Setup**  We conduct qualitative evaluation on NVIDIA Dynamic Scenes, RealEstate10K, nuScenes test set and DAVIS datasets. For NVIDIA Dynamic Scenes, it has ground truth for novel views. We render these novel views based on pose annotations. For datasets lacking annotations, like DAVIS datasets, we generate novel views by randomly adjusting camera angles and positions.

**Results**  As shown in Figure 3, on NVIDIA Dynamic Scenes, our method outperforms PGDVS[†] in both dynamic and static content. For dynamic content, our method can capture object motion

Table 3: **Ablation studies on NVIDIA Dynamic Scenes at** $128 \times 72$ **resolution.** We analyze the effects of various components, focusing on the influence of different model architectures and different loss functions on the overall performance.

| Model | Full Image | | | Dynamic Area | | | Static Area | | |
|---|---|---|---|---|---|---|---|---|---|
| | PSNR↑ | SSIM↑ | LPIPS↓ | PSNR↑ | SSIM↑ | LPIPS↓ | PSNR↑ | SSIM↑ | LPIPS↓ |
| w/o Temporal-based 3D Constraint | 25.78 | 0.830 | 5.60 | 20.71 | 0.691 | 35.1 | 27.34 | 0.857 | 4.84 |
| w/o Self-attention | 27.68 | 0.854 | 5.12 | 21.98 | 0.695 | 34.2 | 28.64 | 0.867 | 4.27 |
| w/o Plane-attention | 27.93 | 0.871 | 4.54 | 23.76 | 0.797 | 16.9 | 29.01 | 0.883 | 2.78 |
| w/o LPIPS Loss | 29.80 | 0.914 | 5.59 | 25.13 | 0.857 | 16.3 | 30.97 | 0.922 | 2.68 |
| w/o SSIM Loss | 27.51 | 0.857 | 4.17 | 23.09 | 0.765 | 18.5 | 28.73 | 0.872 | 2.71 |
| w/o Distortion Loss | 28.43 | 0.888 | 3.72 | 24.12 | 0.822 | 14.9 | 29.58 | 0.899 | 2.30 |
| Ours | 28.56 | 0.884 | 4.25 | 24.27 | 0.810 | 15.0 | 29.78 | 0.896 | 2.47 |

precisely without missing parts, such as the person's head and legs (first column). Besides, our model avoids motion blur for dynamic objects, like person body (second column). For static content, our model achieves superior background modeling without depth priors. Due to limited depth priors, PGDVS[†] performs worse than our method (third and fourth columns), demonstrating the limits of relying on pre-trained priors, such as depth priors.

Figure 4 highlights the flexibility of our framework across diverse scenarios. On nuScenes, it excels in synthesizing novel views for unbounded scenes and dynamic objects like moving vehicles, handling vertical and horizontal viewpoint changes (bottom left). For RealEstate10K, it achieves high-fidelity indoor scene reconstruction with coherence even in unseen domains (bottom right). In the more complex DAVIS scenarios, our framework effectively manages intricate spatio-temporal dynamics, producing smooth and coherent results (top rows).

## 4.2 ABLATION STUDY

In this section, we perform ablation studies on the NVIDIA Dynamic Scenes dataset to better understand the contributions of different components to the efficacy of our approach. Due to computational resource limitations, these studies employ images with a resolution of $128 \times 72$, using a batch size of 32 for 500 epochs throughout the training process.Qualitative results are provided in Appendix A.3 for further reference. Additionally, we explain the static/dynamic detection mechanism of temporal-aware view-attention in detail in Appendix A.2.

**Temporal-based 3D Constraint**   To investigate the impact of our temporal-based 3D constraint , we conduct an experiment that uses only a single view as the target frame during training. The experimental results in Table 3 reveal that temporal-based 3D constraint significantly improves the performance across all metrics. This strategy leverages the disparity between two target views to impose geometric constraints on the generated triplanes, resulting in more accurate multiview consistency. In contrast, the approach based on single target view lacks this constraint and suffers from scale ambiguity, resulting in a noticeable pixel shift in the rendered images.

**Self-attention in Image Encoder**   We investigate the impact of the self-attention module by removing it from the image encoder. The results, detailed in Table 3, show a significant decrease in novel view synthesis metrics, especially in the synthesis of dynamic objects. This decrease is derived from the lack of long context for image encoder. The self-attention can resolve this issue and benefit motion-aware feature aggregation. This demonstrates the critical role of self-attention in the dynamic scene synthesis.

**Plane-attention**   To evaluate the effect of plane-attention, we conduct an ablation study on plane-attention. The results in Table 3 indicate that plane-attention can boost our model on all metrics. This validates that plane-attention has a benefit on the improvement of triplane features.

**Impact of Losses**   As shown in Table 3, without LPIPS loss, our model has an obvious drop on LPIPS metric, showing the effectiveness of LPIPS loss. Besides, we find our model still maintains

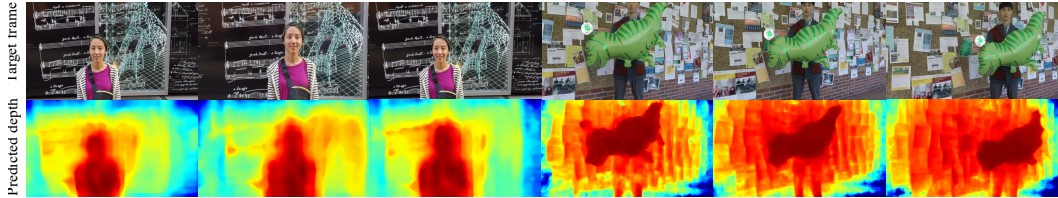

Figure 5: **Reconstructed depth maps on NVIDIA Dynamic Scenes.** Red indicates closer distances, while blue denotes farther distances.

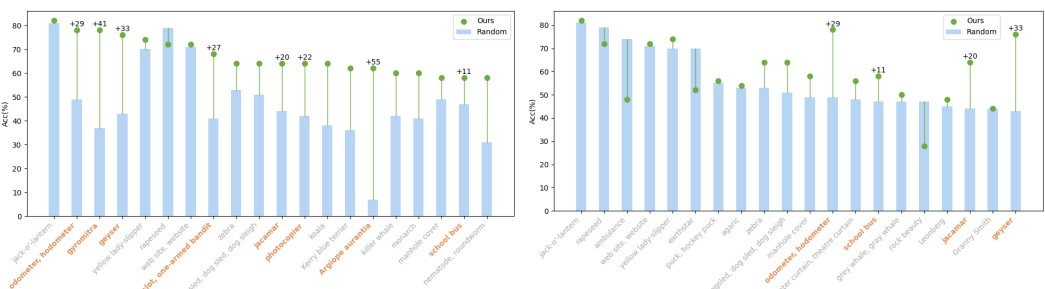

Figure 6: **Comparisons on ImageNet linear classification**: our model vs a random-initialized model. The highlighted categories (orange) are closely related to our training data. The categories in left and right parts are selected by top-1 classification accuracy of our model and a random-initialized model, respectively. We show the top 20 categories.

favorable LPIPS values even without LPIPS, underscoring its inherent capability to capture perceptual quality. For SSIM loss, it boost the performances on PSNR and SSIM, benefitting the learning of low-level high-frequency details. Despite distortion loss has a negative effect on SSIM and LPIPS, it can address the "floater" issue of the rendered image, resulting in a positive gain on PSNR (Full Image).

### 4.3 EMERGENT CAPABILITIES

**Geometry Learning**   We illustrate the depth maps of our model on NVIDIA Dynamic Scenes in Figure 5. The results show that our model can learn to predict depth through self-supervised learning, even without any prior knowledge. Regarding the blocky appearance of the depth map on the right side of Figure 5, we find the blocky areas are closely related to posters. We conjecture that our model can distinguish each individual poster but does not recognize that these posters are on the same wall plane. Although the current depth results are not perfect, this capability is expected to improve as more data becomes available.

**Semantic Learning**   To validate the representation learning of our method, we report the linear probing top-1 accuracy of our image encoder on ImageNet. We use a random-initialized model as the baseline. See detailed experimental settings in Appendix A.4. As shown in Figure 6, the highlighted categories (orange), such as odometer and school bus, are closely related to our training data. Notably, our model significantly outperforms the baseline on these categories, indicating that it has effectively learned semantic information from the training data. This suggests that our method is a potential representation approach when the training data increases.

### 5 CONCLUSIONS

We present a generalizable dynamic radiance filed in first person, which is trained with large-scale monocular videos in a self-supervised manner. Our method can predict the neural representation of physical world conditioned on a sequence of egocentric observations without test-time training. Also, our model can perform dynamic novel view synthesis on seen and unseen scenarios. Moreover, the emergent capabilities show that our method is a potential path to build visual intelligence. We hope our approach can inspire more future research to this task.

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

## A  APPENDIX

### A.1  ARCHITECTURE DETAILS

**Image Encoder**  The image encoder consists of a resnet-like backbone and a self-attention layer. The ResNet component comprises three downsampling layers and nine ResNet blocks. Additionally, the self-attention layer employs 2D sinusoidal position encoding, with a channel dimension of 576.

**Camera Encoder**  The camera encoder consists of a straightforward linear layer. Camera intrinsics are processed through a sinusoidal encoding function before being input into the camera encoder. This linear mapping is designed to align the channel dimensions of the camera features with those of the image features, facilitating consistent feature integration.

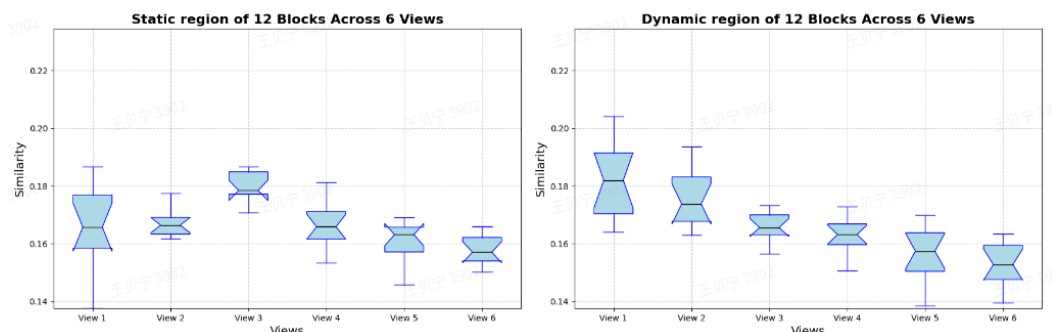

Figure 7: **Boxplot summarizing the similarity measures of dynamic and static 3D points.**The x-axis represents 6 views from the source frames, ordered by increasing temporal distance from the target view.

**4D-aware Transformer**   Feature aggregation is achieved through a stack of 12 basic 4D-aware Transformer modules with an output dimension of 576.

**Upsampler**   The upsample module consists of a single deconvolution layer that upscales the triplane from $32 \times 32 \times 576$ to $128 \times 128 \times 192$. Given the interdependent nature of the planes, we implement an approach adapted from Rodin (Wang et al., 2023a). Specifically, for each pixel in a plane, its feature is concatenated with the average feature of the corresponding row or column from the other two planes, enhancing the contextual integration across the triplane structure.

**Triplane Decoder**   The triplane decoder uses a simple two-layer MLP. Position encoding for the triplane features follows the methodology described in (Wang et al., 2023b).

### A.2   The static/dynamic detect mechanism in Temporal-aware View-attention

**Setup**   We explain the static/dynamic detect mechanism of temporal-aware view-attention using an analysis of a target image from the NVIDIA "playground" scene. Firstly, the pixels from the target image are lifted to 3D points based on predicted depths and classified into static/dynamic points using the semantic mask. Next, the similarities between these 3D points and their corresponding epipolar points from six source views/times are computed across 12 temporal-aware view-attention blocks. Finally, we compute the mean and variance of similarities for static and dynamic points, respectively.

**Results**   As shown in Figure 7, for 3D virtual points, dynamic points show high similarity in close time and low similarity in remote time. In contrast, for static points, the similarity approximates across different times/views. This phenomenon shows that temporal-aware view-attention acquires the ability to differentiate between dynamic and static points via similarity.

### A.3   Qualitative comparison of ablation studies on NVIDIA scenes

We evaluate the impact of temporal-based 3D constraint, self-attention in the image encoder, plane-attention, and the applied loss functions, with the qualitative results presented in Figure 8.

### A.4   Comparisons on ImageNet linear classification

To evaluate semantic learning of our model, we conduct a linear classification experiment on the ImageNet using our image encoder. Note that the encoder is trained without LPIPS loss to avoid semantic leakage. We set randomly initialized image encoder as the baseline. To ensure the stability of the experimental results, each model is trained for 3 times with different seeds and uses the averaged top-1 classification accuracy as the final result.

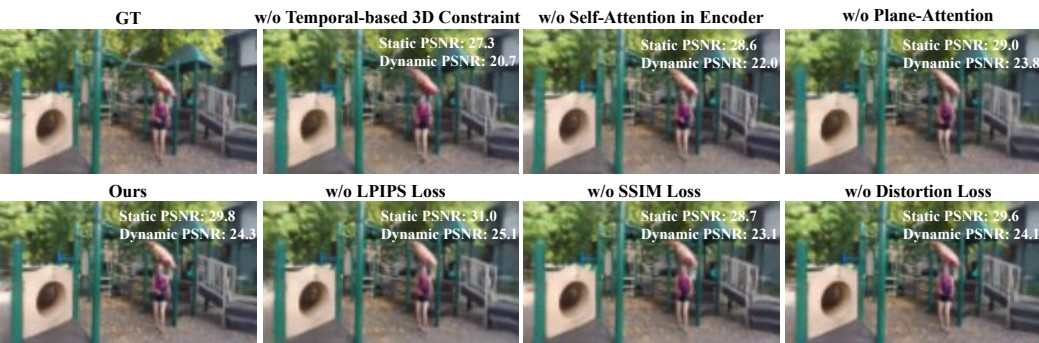

Figure 8: **Qualitative comparison of ablation studies on NVIDIA Dynamic Scenes dataset at** $128 \times 72$ **resolution.** The metrics mean the PSNR on the testing dynamic and static contents in different ablation studies.

## A.5 LIMITATIONS

There are three limitations for our method. First, we need camera intrinsics and extrinsics to train our model. These camera parameters may not align well with the ground truth. Meanwhile, these camera parameters are not easy to obtain for complex videos. For this limit, it is an alternative solution to optimize these camera parameters and our model jointly via a data-driven manner. Second, our model is deterministic, thus it performs not well for unseen content that does not exist in source views. To solve it, we will introduce diffusion models to generate unseen content in the future. Third, due to limited resource, we can not train a large model with large-scale datasets (*e.g.* Ego4D (Grauman et al., 2022)) to validate the scalability of our approach.

