# OpenReview forum: "Generalizable Dynamic Radiance Field in Egocentric View"
_ICLR.cc/2025/Conference — Submitted to ICLR 2025_

### Official Review · Reviewer_aFqB · 2024-10-29

**Soundness:** 2
**Presentation:** 3
**Contribution:** 2
**Rating:** 5
**Confidence:** 5

**Summary:**

This paper studies the generalizable model for dynamic scene rendering. It uses triplane as scene representation, and proposes view-attention, axis-attention and plane-attention modules to optimize the triplane features. Experimental results on RealEstate10K and Nvidia Dynamic Scene datasets show the proposed method achieves comparable results without any finetuning on dynamic scenes compared to other methods that all require per-scene optimization.

**Strengths:**

It could achieve good dynamic scene rendering performance without flow and depth supervision.
It achieves better generalization ability compared to MonoNeRF and PGDVS.

**Weaknesses:**

After carefully reading the method section, I'm still confusing how the proposed method processes dynamic scenes over time. Line 139 says that the model could render novel views at target time $t_1$, but does not mention how $t_1$ is determined and introduced into the model. Figure 1  does not present how the model could generate novel views of dynamic scenes at different timestamps either.

Besides, since this paper uses 3D triplane representation without a time axis and uses all the video frames to train the 3D representation, the time-variant motion features may be mixed into one triplane representation, which is quite confusing to use the model to render novel views at different timestamps.

The paper claimed that it could learn semantic information with the proposed pipeline, but it only tests its image encoder performance on image classification tasks, which is somehow weak. Semantic learning of dynamic scenes should be demonstrated by challenging semantic tasks like segmentation, tracking, completion, and generalization in Semantic Flow [1] paper.

The paper uses many non-egocentric view datasets for evaluation. It is weird to conduct experiments on these dataset as the goal of the paper is to build generalizable dynamic radiance fields in egocentric view. I want to know why.

Reference:
[1] Tian, Fengrui, Yueqi Duan, Angtian Wang, Jianfei Guo, and Shaoyi Du, "Semantic Flow: Learning Semantic Fields of Dynamic Scenes from Monocular Videos", ICLR 2024. https://arxiv.org/abs/2404.05163

**Questions:**

Some citation formats in this paper seem wrong. Please double-check citation formats.

---

> ### Author Response · Authors · 2024-11-20
> **rebuttal**
>
> Thank you for your careful review and helpful suggestions!
>
> **Q1&Q2: How the model could generate novel views of dynamic scenes at different timestamps. The time-variant motion features may be mixed into one triplane representation.**
>
> **A1&A2:** We are sorry to make you confused. We clarify it in revised paper. Concretely, our method is as follows: we use a learned triplane for 3D representation. When rendering novel views at target time $t_1$, we use the learned triplane as the initialization and update the learned triplane by aggregate features from source views at other times using view-attention module. To compute attention scores, we also take time $t_1$ and other times {t}$_{t=1}^S$ as query and key, respectively. In this way, the view-attention module can leverage temporal correlation to aggregate features. Thus, at different times, the triplane feature is updated based on source views and the associated temporal difference.
>
> **Q3: The paper claimed that it could learn semantic information with the proposed pipeline, but it only tests its image encoder performance on image classification tasks, which is somehow weak. Semantic learning of dynamic scenes should be demonstrated by challenging semantic tasks like segmentation, tracking, completion, and generalization in Semantic Flow [1] paper.**
>
> **A3:** For our method, it is a self-supervised approach. Unlike Semantic Flow[1], we do not leverage any semantic labels. As shown in **semantic learning** section, the high classification performance on related categories demonstrates that our method can emerge the capability of semantic learning without any semantic supervision. From this point, our method is a promising approach for representation learning. For challenging tasks in Semantic Flow [1] paper, we think it can be solved by our method when the training data increases. Due to training data issues, we leave it for future work. **We must note that we provide a promising path to learn semantic in self-supervised manner in this paper.**
>
> **Q4: The paper uses many non-egocentric view datasets for evaluation. It is weird to conduct experiments on these dataset as the goal of the paper is to build generalizable dynamic radiance fields in egocentric view. I want to know why.**
>
> **A4:**
>
> **egocentric view**: In general, egocentric view is only a modeling approach. It takes observer as world origin to model dynamic scenes. For each video frame, we use camera center as world origin. **That means under ego-view modeling, all videos can be taken as egocentric datasets.**
>
> **many non-egocentric view datasets for evaluation:** For the evaluation, we need use some specific datasets that have multi-view image and pose annotations.  NVIDIA Dynamic Scenes satisfies the dataset requirement, thus we use it for evaluation.
>
> **Q5: Some citation formats in this paper seem wrong. Please double-check citation formats.**
>
> **A5:** Thanks for your suggestions. We have solved it and updated the paper.

---

> > ### Author Response · Authors · 2024-11-27
> >
> > I just want to follow up on our response. If you have any further questions or need more clarification, please let us know. We look forward to hearing from you.

---

> ### Comment · Reviewer_aFqB · 2024-11-27
>
> Thank you for your detailed response.
>
>
> After carefully reading the rebuttal. I believe that the current problem of this paper is the unclear details of method. Although the revised version of the paper provides more technical details, there is no clear overview of how the model is trained (same point with PbCf). In this way, it is not clear why the propose methods achieve better generalization ability. Besides, I am still confused the propose of the "egocentric" in the title, as there is no clear connection between the "egocentric" word and the model details in the proposed method. "use camera center as world origin" is just a definition of  camera coordinate and can be applied to any other dynamic scene reconstruction models.
>
> I think the contribution of paper the authors try to propose is interesting and valuable. However, I believe that it needs further revision due to these unclear expression in the paper. Hence I keep my rating.

---

### Official Review · Reviewer_D5Cb · 2024-11-03

**Soundness:** 3
**Presentation:** 3
**Contribution:** 2
**Rating:** 5
**Confidence:** 5

**Summary:**

This paper proposes a generalizable dynamic radiance field by using attention across view, axis and plane.

**Strengths:**

Propose a attention-based attention to improve triplane for generalizable task.

**Weaknesses:**

The generalizability is tested on only one dataset. While the main focus is on generalizability, the primary results table (Table 1) does not reflect this aspect.

**Questions:**

Could the algorithm be evaluated on additional datasets to assess its generalizability?

Additionally, in Table 1, it states, "Our model is trained on scenes from NVIDIA, EPIC, Plenoptic, and nuScenes (train set)" (Line 370). Why does the performance remain limited compared to other methods?

Missing reference:
DynPoint: Dynamic Neural Point For View Synthesis
NeuPhysics: Editable Neural Geometry and Physics from Monocular Videos

---

> ### Author Response · Authors · 2024-11-20
> **rebuttal**
>
> Thank you for your careful review and helpful suggestions!
>
> **Q1: The generalizability is tested on only one dataset. While the main focus is on generalizability, the primary results table (Table 1) does not reflect this aspect.**
>
> **A1:** We also evaluate the generalizability of our method on RealEstate 10K, as shown in Table 2. To make generalizability of our method clear, we conduct more experiments and update the experiment section. Specifically, we use PGDVS+ (generalizable version of PGDVS) as baseline. As shown in Table 1, our method outperforms PGDVS+ on all metrics by a large margin. We also provide qualitative comparison in Figure 3. Compared with PGDVS+, our method has superior performance on both dynamic and static scenes.
>
> **Q2: Could the algorithm be evaluated on additional datasets to assess its generalizability?**
>
> **A2:** Thanks for your suggestion. We evaluate our method on DAVIS datasets and show the results in Figure 4. To demonstrate the effectiveness of our method, we also provide the video samples in supplementary material.
>
> **Q3: Additionally, in Table 1, it states, "Our model is trained on scenes from NVIDIA, EPIC, Plenoptic, and nuScenes (train set)" (Line 370). Why does the performance remain limited compared to other methods?**
>
> **A3:**
>
> **Limited performance:** Other methods are scene-specific approaches. They overfit each scene by optimization. It is well known overfitting method is better than generalizable method. **We must point out that our method is a data-driven method and has a potential performance improvement. If we train it with large-scale and diverse scenes, our method can have a better performance and generalize to any scene.**
>
> **Q4: Missing reference: DynPoint: Dynamic Neural Point For View Synthesis NeuPhysics: Editable Neural Geometry and Physics from Monocular Videos**
>
> **A4:** Thanks for your suggestion. We will discuss these works in related works.

---

> > ### Author Response · Authors · 2024-11-27
> >
> > I trust our response covered everything, but if there’s anything else you’d like to discuss, please let us know. Thanks again for your time.

---

### Official Review · Reviewer_fo5V · 2024-11-03

**Soundness:** 2
**Presentation:** 2
**Contribution:** 2
**Rating:** 3
**Confidence:** 4

**Summary:**

The paper proposes a framework for creating generalizable dynamic radiance fields from first-person (egocentric) views, enabling view synthesis of dynamic scenes without test-time training. Using a 4D-aware transformer with dedicated attention modules for understanding temporal, spatial information with regard to the camera parameters, the model aggregates features from monocular videos to form a 3D triplane representation, achieving robust generalization across diverse, unseen scenes.

**Strengths:**

- This paper introduces a 4D transformer, that can be generally used in view synthesis.
- This paper conducted extensive experiments on various datasets, to demonstrate the effectiveness and generalizability of the proposed method.

**Weaknesses:**

- The methodology part is not straightforward to understand.
- The methodology is not aligned to the title or motivation of this paper, focusing on 'egocentric views'
- Experimental results are far from 'comparable' to the previous approaches. I understand that previous algorithms optimize view synthesis scene-wise, so the state-of-the-art performance is not expected. However, showing mid-low performance on PSNR and SSIM metrics in table 3, while calling it 'on-par' is not agreeable.
- This paper includes shallow ablation study. Only module-wise plug-in-plug-out ablation is not enough to fully demonstrate the motivation and effectiveness of the suggested modules. This is where the authors can truly argue that their 4D transformer is actually valid in understanding 4D scene information, even though it lacks performance compared to previous algorithms with scene-specific training.
- Overall qualitative results are not curated well to present the effectiveness of the proposed method.

**Questions:**

- Can you explain what exactly the suggested attention modules in 4D transformer are trying to 'learn'? For example, what is 'axis-attention', implicitly? What does it mean, and how does it help to understand 4D scene?
- Can you explain how the suggested modules are specifically designed for 'egocentric' views?

---

> ### Author Response · Authors · 2024-11-20
> **rebuttal**
>
> Thank you for your careful review and helpful suggestions!
>
> **Q1&Q2: The methodology part is not straightforward to understand. The methodology is not aligned to the title or motivation of this paper, focusing on 'egocentric views'**
>
> **A1&A2:** We update the paper and clarify the methodology part. In our methodology, 'egocentric views' is a modeling approach. It takes observer as world origin to model dynamic scenes. For each video frame, we use camera center as world origin. To implement this modeling, we set triplane center as camera center.
>
> **Q3: Experimental results are far from 'comparable' to the previous approaches.**
>
> **A3:** Thanks for your suggestion. We have revised it in our paper.
>
> **Q4: This paper includes shallow ablation study.**
>
> **A4:** I disagree with this statement. **Only having a strong 4D representation, can our model render high novel views at different time.** Thus, the novel view results in ablation study can reflect the ability of our model on 4D scene modeling.
>
> **Q5: Overall qualitative results are not curated well to present the effectiveness of the proposed method.**
>
> **A5:** Thanks for your suggestion. We have provided new qualitative results in Figure 3. The comparison is between PGDVS+ (generalizable version of PGDVS, baseline) and our method. The results demonstrate that our method beats PGDVS+ on both dynamic and static scenes by a large margin.
>
> **Q6: Can you explain what exactly the suggested attention modules in 4D transformer are trying to 'learn'? For example, what is 'axis-attention', implicitly? What does it mean, and how does it help to understand 4D scene?**
>
> **A6:**
>
> **view-attention module(temporal-aware):** As shown in paper, this module is used to aggregate feature from source views to update triplane. To compute attention scores, we also take time $t_1$ and other times ${t}_{t=1}^S$ as query and key, respectively. In this way, the view-attention module can leverage temporal correlation to aggregate features. Besides, we utilize epipolar priors(3D geometry) to extract image features from source views. Thus, temporal-aware view-attention module has a 4D understanding (temporal+3D geometry) on scene.
>
> **axis-attention:** As shown in paper, axis-attention is proposed to reduce the cost of directly applying self-attention on virtual point features. Due to huge virtual point, the cost of applying self-attention on these point directly is expensive.
>
> **plane-attention module:** As shown in paper, epipolar priors used in temporal-aware view-attention is invalid for dynamic scenes. The virtual point features for dynamic scenes need to refine. Thus, in plane-attention module, we leverage 3D priors to refine. 3D prior is derived from two sources. First, triplane itself is a 3D-structured representation. Second, the rendering is based on triplane and high quality rendering means that triplane has 3D prior knowledge. In the training, triplane should learn these 3D prior knowledge from training dataset.
>
> **Q7: Can you explain how the suggested modules are specifically designed for 'egocentric' views?**
>
> **A7:** In general, egocentric view is only a modeling approach. It takes observer as world origin to model dynamic scenes. For any video frame, we use the camera center as world origin. That means under ego-view modeling, any video can be taken as egocentric datasets. To implement this modeling, we set triplane center as camera center.

---

> > ### Author Response · Authors · 2024-11-27
> >
> > I want to kindly follow up and see if you had any further thoughts or questions on our response. We’re happy to provide any additional information you might need.

---

### Official Review · Reviewer_PbCf · 2024-11-04

**Soundness:** 2
**Presentation:** 2
**Contribution:** 1
**Rating:** 3
**Confidence:** 4

**Summary:**

This paper proposes a generalizable dynamic radiance filed in an ego-centric view. Differing from the common NeRF literature, which requires test-time optimization and object centric-view, the method predicts a neural representation from ego-centric image sequence without test-time training. For this purpose, the paper newly proposes a 4D-aware transformer consisting of a View-Attention Module, Axis-Attention Module, and Plane-Attention Module. Using the method, the paper trains a model on a number of training datasets (NVIDIA Dynamic Scenes, EPIC Fields, and nuScenes) and tests on nuScene (test) and RealEstate 10K. However, the performance does not outperform previous related works, and the method has a number of questions in terms of generalization and dynamic representation.

**Strengths:**

S1. The paper proposes a 4D-aware transformer consisting of a View-Attention Module, an Axis-Attention Module, and a Plane-Attention Module.

**Weaknesses:**

W1. The design philosophy of the proposed framework for generalization.

* The paper conducts generalization experiments by training the triplane on several training datasets (NVIDIA Dynamic Scenes, EPIC Fields, and nuScenes) and testing on nuScene (test) and RealEstate 10K.
However, it is questionable whether the proposed framework is suitable for dynamic radiance field generalization.
Originally, the learnable triplane aimed to learn three feature planes to embed the target scene context and its temporal change.
However, if the target training set gets diverse, the target of the learnable triplane is unclear.
In the current training scenario, what is the learning goal of the learnable triplane, and what do they learn?

* Also, it would be great to discuss whether the current framework is suitable for dynamic radiance field generalizatoin.
The current frameworks seem unsuitable for radiance field generalization to handle totally unseen and out-of-distribution domain data, such as mountains, caves, or endoscopes.

W2. Performance improvement is not significant.

* In novel view generation in both the seen and unseen domains, the proposed method doesn't outperform the pseudo-generalized methods MonoNeRF Tian et al. (2023) and PGDVS Zhao et al. (2024).

* Also, the paper uses nuScene (test) and RealEstate10K datasets as unseen datasets. However, in terms of domain gap, the training set already includes nuScene (training) and common indoor and outdoor scenes. So, the network is already aware of similar structures, such as the common load scenario and indoor building scenario. The domain gap between the training and testing (unseen) dataset is quite small. If the method is truly generalizable, it should be tested with totally out-of-distribution data.

W3. Novelty in terms of generalization and dynamics representation

* The paper insists that it proposed a generalizable dynamic radiance field estimation framework. For this purpose, the paper proposes a new 4D-aware transformer consisting of a View-Attention Module, an Axis-Attention Module, and a Plane-Attention Module. However, their strength in terms of generalization and dynamics representation is unclear.
It would be great to prove the proposed 4D-aware transformer's effectiveness in both generalization and dynamics representation by comparing it to previous methods. For instance, compared to previous dynamic content embedding methods (Li et al. (2023); Tian et al. (2023)), is the View-Attention Module superior to representing dynamic content?

**Questions:**

Please answer the weakness part.

**Details Of Ethics Concerns:**

None.

---

> ### Author Response · Authors · 2024-11-20
> **rebuttal**
>
> Thank you for your careful review and helpful suggestions!
>
> **Q1: However, if the target training set gets diverse, the target of the learnable triplane is unclear. In the current training scenario, what is the learning goal of the learnable triplane, and what do they learn?**
>
> **A1:** Although the learned triplane is shared by all scenes, it is updated by aggregating the image features from source views at different times. Thus, the updated triplane is different for any target scene. In this way, we can use the updated triplane to render high quality novel views.
>  The goal of the learnable triplane is to learn a good initialization, which is useful to aggregate image features.
>
> **Q2: Also, it would be great to discuss whether the current framework is suitable for dynamic radiance field generalizatoin. The current frameworks seem unsuitable for radiance field generalization to handle totally unseen and out-of-distribution domain data, such as mountains, caves, or endoscopes.**
>
> **A2:** The design philosophy of our method is to learn a good 3D representation for a scene at target time $t_1$.
>
> In our framework, we take a learnable triplane as the 3D representation. For generalization, the learnable triplane should be updated according to different scenes and target times.
>
> For a scene at target time $t_1$, we should update the triplane feature by aggregating image features from source views at times {t}$_{t=1}^S$. During the aggregation, we also utilize temporal correlation between target time and source times to compute attention scores. This is the motivation of (temporal-aware) view-attention module.
>
> Due to epipolar priors used in (temporal-aware) view-attention is invalid for dynamic scenes, the virtual point features have much noise. To solve this issue, we propose plane-attention module and axis-attention module to refine these features.  Axis-attention module can reduce the cost of directly applying self-attention on virtual point features. Plane-attention module can leverages 3D priors to remove the noise from virtual point features.
>
> As shown in experiments, the effectiveness of the proposed method is validated by the generalization performance.
>
> **generalization to handle totally unseen and out-of-distribution domain data**
>
> For our method, it is a generalizable approach and can be applied to any scenes. But it is a data-driven method. With limited training scenes, our method can not be applied to scenes that you mentioned, due to lack of related semantic knowledge. **We must note that generalization ability means the ability of assembling learned semantic knowledge when encountering totally unseen objects/scenes.** It is well known that semantic knowledge is learned from large-scale datasets. Thus, our method have a potential generalization when trained with large-scale datasets. We hope the reviewer to think: **if you are required to describe out-of-distribution domain data, like God, what the image of God in your mind is?**
>
> **Q3: In novel view generation in both the seen and unseen domains, the proposed method doesn't outperform the pseudo-generalized methods MonoNeRF Tian et al. (2023) and PGDVS Zhao et al. (2024).**
>
> **A3:** We must point out that our method outperforms the pseudo-generalized methods MonoNeRF Tian et al. (2023), as shown in Table 1.
>
> PGDVS is a scene-specific approach. It overfits each scene by optimization. It is well known overfitting method is better than generalizable method.
>
> **We must point out that our method is a data-driven method and has a potential performance improvement. If we train it with large-scale and diverse scenes, our method can have a better performance and generalize to any scene.**
>
> **Q4: If the method is truly generalizable, it should be tested with totally out-of-distribution data.**
>
> **A4:** We must note that **generalization ability means the ability of assembling learned semantic knowledge when encountering totally unseen objects/scenes.** It is well known that semantic knowledge is learned from large-scale datasets. Unlike existing methods, our method is a data-driven approach and can benefit from large-scale dataset. Our method have a potential generalization when trained with large-scale datasets.
>
> To make generalizability of our method clear, we conduct more experiments and update the experiment section. Specifically, we use PGDVS+ (generalizable version of PGDVS) as baseline. As shown in Table 1, our method outperforms PGDVS+ on all metrics by a large margin. We also provide qualitative comparison in Figure 3. Compared with PGDVS+, our method has superior performance on both dynamic and static scenes.

---

> ### Author Response · Authors · 2024-11-20
> **rebuttal**
>
> **Q5: However, their strength in terms of generalization and dynamics representation is unclear. It would be great to prove the proposed 4D-aware transformer's effectiveness in both generalization and dynamics representation by comparing it to previous methods. For instance, compared to previous dynamic content embedding methods (Li et al. (2023); Tian et al. (2023)), is the View-Attention Module superior to representing dynamic content?**
>
> **A5:** First, the design philosophy of our 4D-aware transformer is shown in A2. Second, **Only having a strong 4D representation and generalization, can our model render high novel views at different time.** Thus, the novel view results in ablation study can reflect the strength of each module.
>
> **Comparison with previous dynamic content embedding methods (Li et al. (2023); Tian et al. (2023))**
>
> We must point out that it is unreasonable for our (temporal-aware) view-attention module to compare with these dynamic content embedding. The motivation and methodolgy are different. Our method is a self-supervised approach, thus can not take priors as input, such as 2D mask and optical flow. But, for Li et al. (2023) and Tian et al.(2023), they rely on 2D mask and optical flow. Moreover, to our knowledge, these priors may have non-negligible errors, which are harmful to dynamic scene modeling, see Figure 3 (new version of paper).

---

> > ### Author Response · Authors · 2024-11-27
> >
> > I hope our response addressed your concerns. If there's anything else you'd like to discuss, we're happy to assist. Thanks again for your time.

---

> ### Comment · Reviewer_PbCf · 2024-11-27
>
> I appreciate the author's detailed response. I carefully read all other reviews and their corresponding responses.
>
> Most reviewer has similar concerns in the definition and experiments regarding generalization (D5Cb,PbCf,Bx6N), presentation issue of the proposed method (aFqB, fo5V), and the learning objective of triplance representation (aFqB, fo5V,PbCf,Bx6N).
> Even after reading all the reviews and responses, the learning objective and the definition of generalization in this paper are unclear to me. Also, as mentioned by Reviewer Bx6N and PbCf, the current experimental setup can be regarded as in-domain distribution (i.e., similar environments were already seen during training), which is far from generalization.
> So, I think the paper needs clarification on writing, problem definition, and problem formulations.
> Therefore, I maintain my initial rating.

---

### Official Review · Reviewer_Bx6N · 2024-11-04

**Soundness:** 2
**Presentation:** 2
**Contribution:** 2
**Rating:** 5
**Confidence:** 4

**Summary:**

The paper describes an approach for novel view rendering given dynamic egocentric video inputs. The approach is feedforward without optimization at inference time, and is trained to generalize to unseen scenes. The approach uses a triplane as scene representation, which is subsequently updated with a transformer to incorporate the input frames as well as camera parameters. Qualitative and quantitative evaluation provides evidence of the performance and generalizability of the approach.

**Strengths:**

Significance: As the authors rightly point out, existing solutions for dynamic NVS generally can not generalize to novel scenes, limiting their practical use scenarios. Therefore, a solution for generalizable, dynamic NVS given first-person video frames can be significant.

Originality: Though none of the building blocks are new, the technical solution of a triplane representation for dynamic NVS and the refinement of the representation with frame features through a "4D-aware transformer" is sound and novel.

Clarity: Though missing some details (explained in the next section), the paper is overall well-written, conveys the main ideas well, and is easy to follow.

Quality: The results are generally consistent with the authors' claim. It is particularly helpful that authors make clear distinctions between dynamic and static scene components, in-domain and out-of-domain samples in the experiments, and also conduct ample ablation studies.

**Weaknesses:**

I think a major weakness is in the evaluation of the approach, particularly regarding generalization.

Results in Sec.4.1.1 are not quite helpful since the approach is behind a few competing methods and the testing scenes are already seen during training. While no per-scene optimization is needed at inference time, it's possible the network can memorize the scenes to some extent via training.

Results in Sec.4.1.2 provide some insights regarding generalization, but are very limited. Table 2 only shows comparisons with single-view methods on a single dataset. To demonstrate generalization capability, the authors should consider:
* Analyzing results across more diverse testing datasets and scenes;
* Comparison with other multi-view approaches (even if they're static or optimization-based);

Also regarding experiments, it would be valuable to show:
* more distinct views from input views to better understand the limitations
* results and comparisons regarding first-person vs other types of samples.

The paper does not have any analysis nor comparisons regarding latency and speed. This would make clear the efficiency advantage over optimization-based approaches.

Some closely related literature on Gaussian Splatting (GS) is not mentioned, namely
* 4D GS:
    - 4D Gaussian Splatting for Real-Time Dynamic Scene Rendering
    - Spacetime Gaussian Feature Splatting for Real-Time Dynamic View Synthesis
    - Motion-aware 3D Gaussian Splatting for Efficient Dynamic Scene Reconstruction
    - 3D Geometry-aware Deformable Gaussian Splatting for Dynamic View Synthesis
    - Dynamic Gaussian Marbles for Novel View Synthesis of Casual Monocular Videos
* Feedforward GS:
    - MVSGaussian, GS-LRM, MVSplat, etc.

Last but not least, the paper lacks clarity on some key concepts that should be better defined, e.g.
* How is the initial triplane learned/initialized?
* The concept of feature "similarity" is used in a few places but is not properly defined.
* "epipolar feature" appears to be simply projected pixel features. If so, the use of term "epipolar" is uncommon and confusing.
* "time bias" on L200 is not defined.

**Questions:**

Please see the suggestions made in the section above, regarding results, analysis, and clarity. Most critically, I think a more thorough experimental analysis of the generalization capability can significantly improve the paper.

---

> ### Author Response · Authors · 2024-11-20
> **rebuttal**
>
> Thank you for your careful review and helpful suggestions!
>
> **Q1: I think a major weakness is in the evaluation of the approach, particularly regarding generalization.**
>
> **A1:** We thank the reviewer for this very helpful and insightful comment!
>
> To make generalizability of our method clear, we conduct more experiments and update the experiment section. Specifically, we use PGDVS+ (generalizable version of PGDVS) as baseline. As shown in Table 1, our method outperforms PGDVS+ on all metrics by a large margin. We also provide qualitative comparison in Figure 3. Compared with PGDVS+, our method has superior performance on both dynamic and static scenes. In addition, we evaluate our method on DAVIS datasets and show the results in Figure 4. To demonstrate the effectiveness of our method, we also provide the video samples in supplementary material.
>
> **Q2: Comparison with other multi-view approaches (even if they're static or optimization-based)**
>
> **A2:** In the paper, our method focus on monocular video setting, which is more challenging than multi-view settings. Thus, it is unfair and unreasonable to compare with other multi-view approaches.
>
> **Q3: more distinct views from input views to better understand the limitations**
>
> **A3:** We have presented the limitations for our method. For more distinct views from input views, the render views have blurred or black areas, which are unseen content that does not exist in source views. For our method, we only take 6 sequent images as source views. It is easy to meet unseen content when rendering more distinct views from input views. To solve it, we will introduce diffusion models to generate unseen content in the future.
>
> **Q4: results and comparisons regarding first-person vs other types of samples.**
>
> **A4:** Thanks for your suggestion. As shown in Introduction, our work focuses on modeling dynamic world in egocentric view to simulate human. Other types of samples is beyond our focus.
>
> **Q5: The paper does not have any analysis nor comparisons regarding latency and speed. This would make clear the efficiency advantage over optimization-based approaches.**
>
> **A5:** We evaluate the inference speed of our method on an L40 machine using NVIDIA scenes. At a resolution of 288x512, our approach renders each target frame in just **15 seconds**. In comparison, the PGDVS method requires **91.4 seconds** per frame (without depth overfitting)—approximately six times longer. This substantial difference in processing speed underscores the efficiency of our method.
>
> **Q6: Some closely related literature on Gaussian Splatting (GS) is not mentioned**
>
> **A6:** Thanks for your suggestion. We will discuss these works in the final version.
>
> **Q7: How is the initial triplane learned/initialized?**
>
> **A7:**  The goal of the learnable triplane is to learn a good initialization, which is useful to aggregate image features. Although the learned triplane is shared by all scenes, it is updated by aggregating the image features from source views at different times. Thus, the updated triplane is different for any target scene.
>
> **Q8: The concept of feature "similarity" is used in a few places but is not properly defined.**
>
> **A8:** Thanks for your suggestion. “similarity" refers to the attention scores computed between 'query' and 'key' elements within each attention module.
>
> **Q9: "epipolar feature" appears to be simply projected pixel features. If so, the use of term "epipolar" is uncommon and confusing.**
>
> **A9:** Thanks for your suggestion. "epipolar feature" indeed refers to projected pixel features. We will revise it in our paper.
>
> **Q10: "time bias" on L200 is not defined.**
>
> **A10:** “time bias” means the target time and source view times. To compute attention scores in (temporal-aware) view attention, we also take time $t_1$ and source times {t}$_{t=1}^S$ as query and key, respectively. In this way, the view-attention module can leverage temporal correlation to aggregate features. We will clarify this term in the final version.

---

> > ### Author Response · Authors · 2024-11-27
> >
> > We hope that our response to your comments was helpful. If there are any further questions or concerns that you feel were not fully addressed, please let us know. We are eager to ensure that all of your concerns are thoroughly resolved. Thank you for your valuable feedback and looking forward to your thoughts.

---

> > ### Comment · Reviewer_Bx6N · 2024-11-27
> >
> > I really appreciate the authors' effort in the rebuttal and the updates to the draft. I think the PGDVS+ baseline is a very welcome addition for the claims regarding generalization.
> >
> > There seem to be some inconsistencies introduced by the updates. NVIDIA Dynamic Scene was part of the training dataset, but is not mentioned as so in the updated draft. If this is indeed the case, it would explain the changes in performance in Tab.1; however, Tab.2 and Tab.3 remain unchanged, which would indicate the model is the same as before.
> >
> > Other reviewers mostly share the overall concerns regarding generalization and performance, and I don't think the new baseline, even if without the inconsistencies, brings substantial enough change to that assessment.
> >
> > I think responses to other concerns are reasonable, though does not improve the draft. I think multi-view NVS is actually closer to the monocular setting (where the frames can be considered as views) compared to the single-image NVS baseslines included in the draft.

---

### Author Response · Authors · 2024-11-20
**Overall Rebuttal**

We thanks all the reviewers for their hard work! We are very honored that our work has been recognized for: **1) sound and novel(Reviewer Bx6N); 2) well-written and easy to follow (Reviewer Bx6N); 3) extensive experiments (Reviewer fo5V); 4) good performance without flow and depth supervision (Reviewer aFqB).**

## Updated Results
To make generalizability of our method clear, we conduct more experiments and update the experiment section. Specifically, we use PGDVS$^{\dagger}$ (generalizable version of PGDVS) as baseline. As shown in Table 1, our method outperforms PGDVS$^{\dagger}$ on all metrics by a large margin. We also provide qualitative comparison in Figure 3. Compared with PGDVS$^{\dagger}$, our method has superior performance on both dynamic and static scenes. In addition, we evaluate our method on DAVIS datasets and show the results in Figure 4. To demonstrate the effectiveness of our method, we also provide the video samples in supplementary material.

Here, we list some critical results to show the effectiveness of our method.

**Full Image**
| Model  | PSNR↑ | SSIM↑ | LPIPS↓ |
|--------|-------|-------|--------|
| PGDVS$^{\dagger}$ | 21.15 | 0.814 | 142.3  |
| Ours   | 22.43 | 0.706 | 16.29  |

**Dynamic Area**
| Model  | PSNR↑ | SSIM↑ | LPIPS↓ |
|--------|-------|-------|--------|
| PGDVS$^{\dagger}$ | 15.93 | 0.479 | 233.5  |
| Ours   | 18.64 | 0.652 | 33.04  |

**Static Area**
| Model  | PSNR↑ | SSIM↑ | LPIPS↓ |
|--------|-------|-------|--------|
| PGDVS$^{\dagger}$ | 23.36 | 0.854 | 129.9  |
| Ours   | 24.03 | 0.724 | 15.79  |

---

### Author Response · Authors · 2024-12-02

Dear Reviewers,

Thank you for your insightful feedback and for recognizing the potential contributions of our work.

In response to your comments, we will refine the problem definition of our method to enhance the clarity and readability of our paper. Additionally, we are committed to strengthening our experimental design to more effectively validate our findings.

Best regards,

Authors

---

### Meta-Review · Area_Chair_TMhE · 2024-12-23

**Metareview:**

This paper proposes a framework for constructing a generalizable dynamic radiance field in a first-person view in a feed forward manner. To achieve this the authors encode the initial scene into a triplane and then subsequently update it with a 4D-aware transformer module. The authors train on several datasets including several non-ego centric ones and evaluate performance on two held-out datasets. The experimental results show some evidence of the performance and generalizability of the proposed approach.

**Additional Comments On Reviewer Discussion:**

Five reviewers provided the final scores of 5, 3, 3, 5, 5 for this work. Several reviewers (D5Cb,PbCf,Bx6N) raised concerns about the generalization of this work; lack of comparisons to other state of the art methods; experimental setting; the presentation of its method (aFqB, fo5V) and its learning objective (aFqB, fo5V,PbCf,Bx6N). The reviewers additionally raised concerns about the claim of the method being applicable to ego-centric views, while the experiments were conducted largely on non-ego centric datasets. In response to the reviewers' concerns the authors provided additional comparisons to one other generalizable method PGDVS method, However, these updates did not sufficiently address the reviewers' concerns and they unanimously voted to reject this work.

The AC agrees with the assessment of the reviewers and hence recommends rejection.

---

### Decision · Program_Chairs · 2025-01-22

Reject